# An Advanced Protocol for the Quantification of Marine Sediment Viruses via Flow Cytometry

**DOI:** 10.3390/v13010102

**Published:** 2021-01-13

**Authors:** Mara Elena Heinrichs, Daniele De Corte, Bert Engelen, Donald Pan

**Affiliations:** 1Institute for Chemistry and Biology of the Marine Environment, Carl von Ossietzky University of Oldenburg, 26129 Oldenburg, Germany; mara.elena.heinrichs@uni-oldenburg.de (M.E.H.); daniele.deco@gmail.com (D.D.C.); engelen@icbm.de (B.E.); 2Institute for Extra-Cutting-Edge Science and Technology Avant-Garde Research (X-Star), Japan Agency for Marine-Earth Science and Technology (JAMSTEC), Yokosuka 237-0061, Japan; 3Department of Ecology and Environmental Studies, The Water School, Florida Gulf Coast University, Fort Myers, FL 33913, USA

**Keywords:** flow cytometry, Nycodenz, density gradient centrifugation

## Abstract

Viruses are highly abundant, diverse, and active components of marine environments. Flow cytometry has helped to increase the understanding of their impact on shaping microbial communities and biogeochemical cycles in the pelagic zone. However, to date, flow cytometric quantification of sediment viruses is still hindered by interference from the sediment matrix. Here, we developed a protocol for the enumeration of marine sediment viruses by flow cytometry based on separation of viruses from sediment particles using a Nycodenz density gradient. Results indicated that there was sufficient removal of background interference to allow for flow cytometric quantification. Applying this new protocol to deep-sea and tidal-flat samples, viral abundances enumerated by flow cytometry correlated well (R^2^ = 0.899) with counts assessed by epifluorescence microscopy over several orders of magnitude from marine sediments of various compositions. Further optimization may be needed for sediments with low biomass or high organic content. Overall, the new protocol enables fast and accurate quantification of marine sediment viruses, and opens up the options for virus sorting, targeted viromics, and single-virus sequencing.

## 1. Introduction

Viral particles are highly abundant in marine sediments and typically range between 10^7^ and 10^10^ per g of dry sediment [1], which exceeds bacterial cells. With most assumed to be bacteriophages [2], viruses are recognized as playing a key role in mediating microbially driven biogeochemical cycles and ecosystem processes by controlling microbial mortality, diversity, and evolution in the environment [3,4,5,6]. While knowledge of planktonic viruses has grown considerably in recent decades, there are still many questions regarding the function of viruses in marine sediments due to the difficulty of sample recovery and the analytical challenges associated with the sediment matrix.

In general, viral abundance is a basic parameter for understanding virus-host interactions and the dynamics of viruses in sediments. Epifluorescence microscopy (EfM) has long been a standard technique for viral enumeration in aquatic environments [7,8]. Recently, flow cytometry (FCM) has emerged as a common method for the quantification of aquatic viruses [9,10]. Compared to EfM, FCM analysis is fast and allows a high sample throughput. However, flow cytometric enumeration of sediment viruses is still challenging. To date, FCM has been tested in a few studies for the analysis of sediment viruses. Owing to their small size, most viruses are close to the detection limit of the FCM and can overlap with instrumental noise or background fluorescence deriving from non-viral particles. The presence of this interference can lead to inaccurate virus counts [11,12]. Furthermore, larger sediment grains and particles can clog the instrument nozzle [13]. Therefore, efficient extraction of viruses from the sediment matrix is necessary to minimize the problems associated with the presence of non-biological particles [14,15].

Conventional sediment virus extraction protocols typically involve a combination of physical and/or chemical treatments, followed by centrifugation and filtration to remove cells and larger particles. With these protocols, very few cases of successful application of flow cytometry to sediments or soil have appeared in the literature. One study for example demonstrated the potential of flow cytometry for the enumeration of viruses from freshwater lake sediment [16]. However, only two sediment types with similar compositions were analyzed in this study. Williamson et al. [17] attempted flow cytometry on a range of different soils, but found poor correlations between virus abundances determined by EfM and FCM, with counts varying by as much as three orders of magnitude. High background fluorescence in some soil samples was identified as the cause for these deviations, indicating the insufficient removal of interfering particles. This was also reported by Brussard et al. [10] for different sediment types. The efficiency of virus extraction may also vary with sediment composition, as the strength of adsorption depends on the mineral surface and virus particle properties [18]. Clays, for example, negatively influence virus recovery due to the electrostatic interactions that strongly bind virus particles [19,20,21]. Thus, the results of virus counts can deviate greatly, making it hard to compare results between studies utilizing different extraction methods or even within the same study between samples of different compositions [17,22].

Previously, a buoyant density-based method was developed for the extraction of microbial cells from sediments, which reduced interference associated with sediment particles [13,23]. The method utilizes a Nycodenz gradient step for the separation of cells from sediment particles. The extraction process was efficient enough to allow for the enumeration of cells by FCM [13]. Recently, the method was extended to improve extraction and enumeration of sediment viruses using EfM [22]. In our study, we tested the applicability of this method for counting marine sediment viruses via FCM. We tested this method on a variety of sediment types using deep-sea and tidal-flat sediments of different compositions and lithological characteristics. The FCM counts were compared to EfM numbers following the same extraction procedure. To account for lab and instrument variability, we verified the method with two different flow cytometers in two independent labs.

## 2. Materials and Methods

Here, viral particles were detected as SYBR Green I-stainable particles that pass through 0.2 µm pore size filters. This criterion excludes a certain fraction of viruses, for example large viruses, prophages or viruses that are not efficiently stained by SYBR Green I (e.g., RNA and ssDNA viruses). On the other hand, non-viral particles such as gene transfer agents, membrane vesicles, cell debris or small cells that pass through 0.2 µm filters may also be erroneously detected as viral particles [24,25]. Keeping this uncertainty in mind when quantifying viruses via fluorescent staining, we subsequently inferred viral abundances and viral counts based on this criterion.

### 2.1. Sample Inventory

A variety of sediment types, both deep-sea and tidal-flat sediments, from different biogeochemical and lithological backgrounds was chosen for analysis by FCM and EfM (Table 1). The deep-sea sediments were sampled at four individual sites in the Pacific Ocean with different environmental characteristics (e.g., productivity, sedimentation rate, and biogeochemical patterns). The sediments were retrieved by using a multicorer (Octopus, Kiel, Germany) along a transect spanning from 40° S to 58° N close to the 180° meridian during two consecutive research cruises aboard RV Sonne (expeditions SO248 and SO254). Detailed sample retrieval and preparation are described in Heinrichs et al. [26]. In brief, sediment slurries (mixed 1:1 with artificial seawater) were prepared from sediment cores sampled from the upper 20 cm of the seafloor (water depths between 3000 and >5000 m). For enumeration of the virus particles, 0.5 mL subsamples of the slurries were flash-frozen using liquid nitrogen and kept at −80 °C until analysis.

Additionally, fresh tidal-flat sediments from the German Wadden Sea were sampled at the beach of Dangast (North Sea, German Bight). Two sample types with predominantly sandy and muddy sediment composition were collected near and below the low-water line of the beach to a depth of 10 cm using sterile spatulas. The samples were stored in the dark at an in situ temperature of 4 °C for two weeks until analysis. Before extraction, slurries were prepared by mixing the tidal-flat sediments 1:1 with 0.02µm-filtered 2.5% sodium chloride.

### 2.2. Geochemical Characterization of Sediments

Detailed descriptions of the geochemical analyses of the deep-sea sediments are found in Pohlner et al. [27]. In brief, chlorophyll a concentrations were recorded with a fluorometer mounted on a CTD rosette during the cruises. Here, the total chlorophyll a measured in the top 500 m of the water column served as an estimate for the productivity of the overlying water column. Sedimentary total organic carbon (TOC) content was analyzed by subtracting the inorganic carbon fraction from total carbon, both measured with an elemental analyzer (Eltra CS-800, equipped with an acidification module Eltra CS-580, Haan, Germany). Samples for the analysis of Si and Al were freeze-dried, homogenized, fused to glass beads as described by Zindorf et al. [28], and afterwards analyzed by wavelength dispersive X-ray fluorescence (Axios FAST, Malvern Pananalytical, Malvern, UK). The ratio between these major elements was used as a proxy for the clay content of the sediment. The grain size distributions of the <2000 µm fraction of tidal-flat sediments were determined using a laser diffraction spectrometer (Horiba LA-950, Kyoto, Japan). Three subsamples were measured as a suspension, omitting pretreatment with hydrogen peroxide or hydrochloric acid.

### 2.3. Extraction of Viruses

To remove particles that may interfere with FCM and EfM analysis, viruses were separated from the sediment matrix utilizing pyrophosphate, sonication, and a Nycodenz density gradient as described by Pan et al. [22]. A schematic overview of the extraction procedure is given in Figure 1. A more detailed user protocol for the extraction and analysis of sediment viruses via FCM is provided in the Appendix A. Briefly, sediment samples were thawed on ice (if frozen) and mixed with a solution of 2.5% sodium chloride and 5 mM pyrophosphate to create slurries. To test the extraction method over a wide range of virus abundances, tidal-flat sediment slurries were analyzed at various dilutions (1:2, 1:10, 1:100, 1:1000). Thereby, each dilution was prepared from an independent subsample before extraction. Samples were sonicated 3 times on ice (1 min, with a 30 s pause between each round of sonication) to detach viruses from the sediment. The sonicated samples were then layered onto a two-step gradient of 30% and 50% Nycodenz (Axis Shield, Dundee, UK) and centrifuged at low speed (30 min at 2900× *g*). This procedure limits the readsorption of viruses back onto sediment particles after separation. Larger and denser particles migrate to the denser Nycodenz layer, while the majority of viruses remains in the top and 30% Nycodenz layer. Some viruses still associated to particles may cross the 50% layer and are recovered in a second round of extraction. The virus-containing fraction was filtered through 0.2 µm filters (PES membrane, Sartorius, Göttingen, Germany) to further remove cells and particles. The extraction was performed twice for each sample and both extracts were combined. A subsequent DNase I treatment (final concentration 0.5 U/mL, Thermo Fisher Scientific, Waltham, MA, USA) reduced the background fluorescence of extracellular DNA as suggested by Danovaro and Middleboe [15]. For FCM analysis, three-quarters of the virus-extract (approximately 9 mL) was concentrated down to 500 µL by Amicon filters (100 kD, Merck Millipore, Burlington, MA, USA) and washed with the 2.5% sodium chloride solution. This step was included to minimize carryover of Nycodenz, which may potentially influence the laminar flow of particles within the flow cell. Additionally, this pre-concentration step is necessary in order to yield sufficiently high concentrations for flow cytometric analysis, as samples are recommended to be diluted at least 10-fold in TE buffer prior to analysis [10]. The remaining virus-extract was used for enumeration by EfM. Both extracts were fixed with electron microscopy-grade glutaraldehyde (0.5% final concentration), incubated at 4 °C for 30 min, and stored at −80 °C until analysis. Autoclaved tidal-flat sediments and 0.02 µm-filtered Milli-Q water served as extraction blanks and negative controls.

### 2.4. Flow Cytometry

Extracts were prepared and analyzed according to Brussaard et al. [10]. First, samples were diluted with 0.02 µm-filtered TE buffer (pH 8.0, 10 mM Tris, 1 mM EDTA, Sigma Aldrich, St. Louis, MO, USA), stained with SYBR Green I (0.5 × final concentration, Thermo Fisher Scientific, Waltham, MA, USA), incubated at 80 °C for 10 min, and cooled for 5 min at room temperature. Samples with very low fluorescence were stained with a final SYBR Green I concentration of 2× to improve viral detection. TE blanks were prepared and analyzed in an identical manner as the samples and run between every sixth sample to assess instrument noise and minimize sample carryover. To verify this new protocol, extraction and enumeration was performed in two different labs and on two different instruments. One set of sediment samples was analyzed using an Attune NxT flow cytometer (Thermo Fisher Scientific, Waltham, MA, USA)) at the Japan Agency for Marine Earth Science and Technology (JAMSTEC). Another set was prepared and measured on an Accuri C6 flow cytometer (Becton Dickinson, Franklin Lakes, NJ, USA) at the Institute for Chemistry and Biology of the Marine Environment (ICBM) of the University of Oldenburg, Germany. Both flow cytometers are equipped with a blue laser at 488 nm. The fluorescence threshold of the Accuri instrument was set on FL-1:550 after assessing the background noise of the instrument. Samples were analyzed on both flow cytometers, maintaining an event rate between 200 and 800 events per s to avoid coincident particle detections. Viral abundance was calculated after background correction using TE blanks. Data was processed in FlowJo v.10 (FlowJo LLC, Becton Dickinson, Franklin Lakes, NJ, USA).

### 2.5. Epifluorescence Microscopy

To verify quantification by FCM, virus enumeration was also conducted using EfM, following Suttle and Fuhrman [29]. Briefly, virus extracts were thawed at room temperature and diluted with 0.02 µm-filtered PBS buffer to yield approximately 20–50 virus particles per field of view. Samples were then vacuum-filtered onto 0.02 µm Anodisc filters (Whatman, Maidstone, UK) and stained with SYBR Green I (20 × final concentration) for 15 min in the dark. After removal of excess stain, filters were mounted onto microscopic slides. At JAMSTEC, slides were mounted with Vectashield antifade mounting medium (H-1000, Vector Laboratories, Burlingame, CA, USA) and enumerated at 1000× magnification on an Olympus BX53 microscope (130 W U-HGLGPS fluorescence light source, excitation filter 460–480 nm, emission filter 495–540 nm, Tokio, Japan). At ICBM, slides were mounted with 0.1% p-phenylenediamine antifade solution [29] and enumerated at 1000× magnification using a Leica DM RBE microscope (Leica, Wetzlar, Germany), equipped with a 50 W HBO AC-L1 mercury lamp (excitation wavelength 450–490 nm, cut-off filter 515 nm). In both cases, ten to twenty random fields were captured by camera and at least 300 virus particles per sample were counted using ImageJ (version 1.52k). Statistical analyses were performed using RStudio (version 1.1.383).

## 3. Results

A set of marine sediments with a variety of different lithological characteristics was selected to test the method across different sediment types. The sediments can roughly be subdivided into shallow, tidal-flat sediments and deep-sea muds from different geographical areas. The geochemical analyses of the deep-sea sediments revealed differences in the productivity of the overlying water column, sedimentary TOC contents, as well as mineralogical features indicated by the Si/Al ratio (Table 1). The sediment composition, biogeochemistry and hydrography of the tidal-flat site have been described previously [30,31,32]. The TOC content of sediments near the sampling site were reported to be between 2% and 4% [33]. Both sandy and muddy sediments were collected from the tidal flats. The sandy tidal-flat sediments consisted predominantly (60%) of particles in the size range of sand grains (>63 µm), whereas approximately 80% of the grain size distribution of the muddy tidal-flat sediments consisted of particles smaller than 63 µm.

### 3.1. Enumeration of Marine Sediment Viruses by Flow Cytometry

The Nycodenz method has been shown to improve virus separation and recovery from sediment [22]. For the sake of comparing the Nycodenz to a conventional extraction method [15], viruses were extracted from a sediment sample collected from Tarama Knoll (Okinawa Trough [34]) by both methods and then analyzed on the Attune NxT flow cytometer. The Nycodenz separation resulted in greater fluorescence of particles compared to the conventional extraction method. The number of detected particles in the virus gate of the Nycodenz extracted sample was two orders of magnitude higher (2.2 × 10^7^ virus particles per mL) compared to the conventional method (2.3 × 10^5^ per mL) (Figure 2). The flow cytogram of the conventional extraction method showed a dominance by background noise. EfM also indicated that the Nycodenz method improves the fluorescence signal to background ratio, likely a result of the greater separation of interfering particles.

The Nycodenz separation method was tested on two sample types, covering a wide range of virus abundances. Tidal-flat sediments resulted in the highest virus counts, while deep-sea sediments contained the lowest numbers. Calculating the original virus numbers from FCM quantification, virus particles in the muddy tidal-flat samples ranged from 5.9 × 10^8^ to 5.7 × 10^9^ per cm^3^ of sediment and from 1.4 × 10^8^ to 3.4 × 10^9^ in the sandy tidal-flat samples. Viral numbers in the deep-sea samples varied between 4.6 × 10^6^ and 1.9 × 10^8^ virus particles per cm^3^ of sediment for the South Pacific Gyre, between 2.9 × 10^7^ and 5.3 × 10^8^ for the Equatorial upwelling region and between 9.2 × 10^6^ and 2.1 × 10^7^ for the North Pacific Gyre.

Prior to analysis of the samples, gating of virus populations within the cytograms was decided based on comparing samples containing natural virus populations to blanks and autoclaved sediment controls. The virus population gate was close to the detection limit of the instrument for all samples (Figure 3). Autoclaved tidal-flat sediment served to determine gate cutoffs for noise deriving from sediment debris. Some debris in the size range of viruses were still detected as events in the virus gates, on average accounting for 1.3% of the events detected in the corresponding samples containing viruses. Using EfM, false positive counts were on average 2.9% of the virus particles detected in the samples containing the natural virus population. In the 0.02 µm-filtered MilliQ blank control, no identifiable virus population was detected, indicating the cleanliness of the reagents and extraction procedure.

### 3.2. Comparison of Flow Cytometry and Epifluorescence Microscopy

We assessed the FCM method against EfM using a variety of marine sediment samples from the deep sea and tidal flats (Figure 4). Viral abundances ranged between 3.9 × 10^4^ and 1.2 × 10^8^ per ml of slurry, when determined by FCM and between 2.7 × 10^5^ and 2.0 × 10^8^ per ml of slurry, when enumerated by EfM. Linear regression analysis of all samples (*n* = 31) showed that results from both quantification methods were highly correlated (R^2^ = 0.899, *p* < 0.005), indicating that FCM is as suitable as EfM for the enumeration of sediment viruses over the range of tested marine sediments. Lab and instrument variation was low, as approximately two thirds of all data points fell inside the 99% confidence interval of the regression, irrespective of the use of different instruments in individual labs (Table 2). A stronger correlation was observed for the Accuri C6 at ICBM (R^2^ = 0.92) than the Attune NxT at JAMSTEC (R^2^ = 0.71). However, this was likely due to a smaller sample set size and lower abundance range analyzed on the Attune NxT. Correlation between FCM and EfM was strong even among different types of samples and composition, suggesting the suitability of the extraction method for other sediment types.

The counts determined by FCM and EfM differed on average by a factor of 2.7 (range 1–16). For the majority of samples (65%), counts differed by a factor of less than 3. In lower ranges (<10^6^ virus particles per ml of slurry), counts of EfM tended to be higher than those determined by FCM. Within this range, abundances assessed by both counting methods differed on average by a factor of 8.2 (range 4.2–16). Outside of these low abundance samples, the counts differed on average only by a factor of 1.8 (range 1–2.8). Samples measured via FCM exceeded those from EfM by 1 to 7.5 times (on average 2.5), a typical range reported in other studies [16].

## 4. Discussion

### 4.1. The Extraction Method is Suitable for Flow Cytometric Quantification

We measured a strong correlation (R^2^ = 0.899) between virus counts conducted via FCM and EfM, indicating that the method proposed here is suitable for the quantification of marine sediment viruses. The correlation is in the same range as the FCM protocol optimized for aquatic samples [10]. Although the use of FCM has become commonplace for aquatic environments, to date, FCM is still not commonly used for counting sediment-associated viruses due to methodological limitations such as insufficient removal of interfering particles [35]. While some authors have reported the usage of FCM for soil-associated viruses, the protocols have largely been optimized for limited applications such as microbial mats [36] (R^2^ = 0.74) or activated sludge flocs [37] (R^2^ = 0.77). The use of the conventional method for extraction of sediment viruses [15] results in high background levels, which interfere too much with the flow cytometric detection of viruses [10]. By applying our method, we largely overcame this problem through the inclusion of a Nycodenz density gradient separation step (Figure 2), enabling the subsequent analysis of extracted viruses via FCM.

We tested the flow cytometry protocol on a variety of sediment types with different characteristics (Table 1). Based on the combined and individual regression analyses of the samples, we conclude that flow cytometry performed acceptably well for the types of sediments tested in this study (Table 2). The virus abundances we measured for the various sediment types are consistent with other reported values in similar marine environments. For example, values for sandy tidal-flat sediments (1.4 × 10^8^–5.7 × 10^9^ viral particles per cm^3^) resemble those from a study on intertidal sand-flat sediments from the German Wadden Sea not far from our sampling station [38], suggesting there was not a substantial loss of viruses during storage before sample processing. Our numbers for deep-sea surface sediments from the South Pacific Gyre (9.6 × 10^6^–1.9 × 10^8^ virus particles per cm^3^) were similar to those counted from the same region by Engelhardt et al. [39]. The difference of one to three orders of magnitude between tidal-flat and deep-sea sediment viral abundances is in accordance with expectations. As tidal-flat sediments are constantly supplied with fresh substrate, they harbor highly metabolically active microbial communities and therefore support higher cell and virus abundances [30,40]. In contrast, the deep sea is a more energy-limited habitat with reduced biological productivity, activity, and low abundances [41].

### 4.2. Methodological Considerations

For reliable comparisons of viral abundances between samples from different sites, an extraction protocol that works for a variety of sediment types is necessary. Sediment of different origins and compositions differ in physical and chemical structure as well as overall microbial activity. Adhesion forces between viruses and charged surfaces of sediment minerals affect the desorption of viruses [17,18]. Therefore, the efficiency of virus extraction is influenced by sediment properties as viruses vary in their strength of adsorption onto different sediment mineral components, depending on the prevalent ionic strength [19,20]. Virus recovery from sediments with higher clay content is for example less efficient [21]. However, virus extraction protocols may differ widely in efficiency depending on sediment characteristics [22], potentially influencing the interpretation of abundance and distribution data for sediment viruses. Furthermore, particulates also interfere with accurate enumeration for both EfM and FCM. The use of a Nycodenz separation step during virus extraction was previously reported to enable higher yields and more consistent extraction efficiencies from different types of subseafloor sediments [22]. The Nycodenz layer may also act as a physical barrier, preventing re-attachment of viruses onto separated particles. As we only tested marine sediments, this protocol may need to be adjusted to other sediment types such as freshwater, estuarine or more eutrophic sediments.

As viruses and organic matter share similar properties, it may be challenging to separate viruses from dissolved and particulate organics in some organic-rich sediments. Since tidal-flat sediments such as the ones used in this study are high in organic matter [33], this material may not have been sufficiently removed during the extraction procedure (Figure 5). Following Nycodenz separation, the color of the virus-extract from some tidal-flat samples was light yellow-brown, suggesting that some organic components remained in the virus extract (Figure 5). If the volume of organic-rich sediment used for extraction is high, organics may potentially interfere with the detection of viruses. Autoclaved sediment controls demonstrated that some background fluorescence deriving from the sediment extracts falls in the range of viruses (Figure 3). However, these background events are two orders of magnitude lower than the virus signal measured in corresponding non-autoclaved sediment samples. Because of this sediment-derived background fluorescence, the organic load of the sediments may shift the detection limit of the flow cytometer. In lower biomass sediments, it may be more difficult to distinguish the virus signal from interfering background noise. In practice, this might not be a problem for shallow sediments, as they rarely harbor virus numbers below 10^6^ per cm^3^ [1]. However, deep subseafloor sediments tend to have much lower viral abundances [22,39], so further developments may be necessary for flow cytometric enumeration of deep, low biomass samples. It may be possible to achieve a lower detection limit with organic-poor sediments. Samples with high clay, humic content, or high organic matter, such as highly eutrophic sediments, may benefit from extractions using smaller sediment volumes, e.g., by higher dilution of slurries [22,42]. However, the level of dilution that minimizes background fluorescence while maximizing a virus concentration that is still detectable by FCM should be optimized for different types of sediments.

In some of our samples, the viral abundances assessed by EfM exceeded FCM counts, especially in the lower abundance range (<10^6^ virus particles per mL of slurry). In aquatic studies, FCM counts are generally found to be higher than those obtained from EfM [16,36,43,44]. For FCM, reduced quenching of the green fluorescent signal [36] and shorter contact time of fluorescent particles with the laser beam compared to microscopic counting have been given as possible explanations [44]. However, we were careful to minimize fading time of the stain by taking images of virus filters by a camera rather than counting by eye. It is generally advisable to analyze a sub-set of samples via EfM using a camera as a reference method for the flow cytometric counts. Another possible reason why EfM counts may have exceeded FCM numbers in the low abundance samples is the TE buffer dilution of the samples prior to FCM analysis. As low abundance samples cannot be diluted sufficiently with TE before they become too dilute to enumerate via FCM, the pH of the sample may be outside of the optimal range for efficient SYBR Green I staining [45]. Diluting the samples at least 30-fold with TE buffer, or alternatively, increasing the pH of TE to 8.2 as recommended by Mojica et al. [45] may circumvent this problem.

Occasionally, different virus clusters can be distinguished in cytograms of aquatic samples based on their SYBR Green I fluorescence intensity and light scatter [10,46,47]. In our samples, the virus clouds were quite close to the electronic noise of both instruments, and different virus clusters could not be distinguished in the cytograms of any sample (Figure 3). Flow cytometers may differ in their ability to discern distinct viral clusters. However, there are only a few studies with published cytograms of sediment viruses, making it difficult to draw conclusions about the ability to detect distinguishable viral populations. One study identified two virus clusters with different fluorescence intensities in samples extracted from microbial mats [36]. Another study identified three subpopulations following Brussaard et al. [10], with the majority of viruses belonging to the lowest fluorescence gate [37]. However, no clear separation between the individual populations was visible in the published cytograms of Brown et al. [37]. The virus extraction procedure itself may also contribute to the comparably low fluorescence or uniformity of the detected particles, by damaging the virus capsids. Similarly, Amalfitano et al. [11] found that the detachment of microbial cells from sediment particles negatively affects their cell membrane integrity. Regardless of the reason, SYBR Green I concentration can be adjusted for more efficient staining to improve viral detection, e.g., for low fluorescence viruses. It is advisable to process TE blanks using the same SYBR Green I concentration as the samples.

Other density gradient compounds may potentially be substituted for Nycodenz; however we did not test them in this study. One reason Nycodenz was chosen is that it has been demonstrated to work for the separation of cells from subseafloor sediments [13,23,48]. Other density gradient compounds have previously been used for the same purpose, for example, Histodenz [42] and Percoll [49,50,51]. Histodenz has been shown to efficiently remove sediment and soil particles, although there are also challenges with organic matter-rich sediments interfering with cell counts [42]. In a study comparing the density gradients of Percoll and Nycodenz, the latter was found to be superior in the removal of soil particles [51]. However, a distinct decrease in sediment and detritus particles was achieved by using Percoll, but the strong fluorescence of this material influenced counting [49].

### 4.3. Performance on Different Flow Cytometers

We were able to apply the described protocol in two independent labs, utilizing two different flow cytometers (Attune NxT and Accuri C6). This indicates that the method is reproducible for processing and analysis of sediment samples in different labs and on different instruments. Although the regression line correlating FCM and EfM was slightly different for both flow cytometers, the differences in the slopes and intercepts of the regression lines are statistically insignificant (analysis of covariance, *p* < 0.05). However, the sensitivity of virus detection may still vary among different flow cytometers as stained viruses are generally close to the detection limits of the instruments. On the Accuri C6, we were able to detect as few as 3.9 × 10^4^ per mL of slurry. We did not test the lower limits for the Attune NxT due to lack of samples. The performance of other flow cytometers has not been tested, however many modern benchtop flow cytometers have been utilized for enumerating aquatic viruses, and we expect that the method could be translatable to other flow cytometer models and even cell sorters. More recently, sorting of virus particles coupled with multiple displacement amplification (MDA) has been successfully used to study both viral populations (targeted viromics) and/or single viral genomes [52,53,54]. The possibility of assessing sediment viruses with FCM suggests the potential to combine our separation technique with sorting and MDA to investigate viral community composition in much greater detail.

## 5. Conclusions

Overall, the sediment virus separation method using a Nycodenz gradient resulted in less background noise for a range of marine sediment samples, and enabled the fast and reliable quantification of marine sediment viruses via FCM. Since the removal of particles worked so well, the protocol could potentially be optimized for a broader range of sediment types that were not tested here, such as freshwater or estuarine sediments. As challenges with sediments containing a high organic matter load remain, the method should be further optimized before FCM is used for enumerating viruses in organic-rich matrices such as soils or highly eutrophic sediments. However, with the recent introduction of virus sorting and single-virus sequencing for aquatic environments, our method may contribute one step towards the application of these next generation flow cytometric methods to sediment samples.

## Figures and Tables

**Figure 1 viruses-13-00102-f001:**
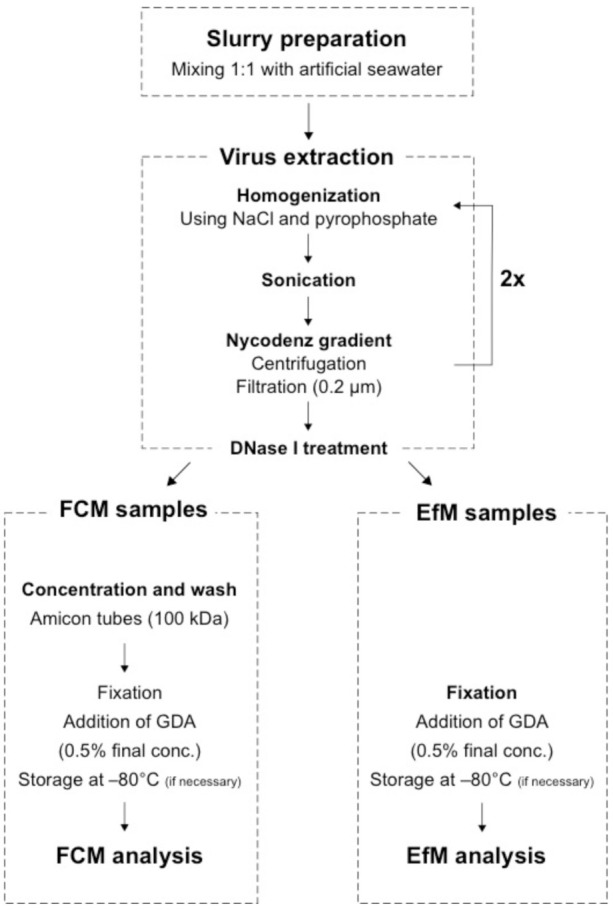
Schematic flow chart of the protocols for flow cytometric and epifluorescence microscopy enumeration. FCM: flow cytometry, EfM: epifluorescence microscopy, GDA: glutaraldehyde.

**Figure 2 viruses-13-00102-f002:**
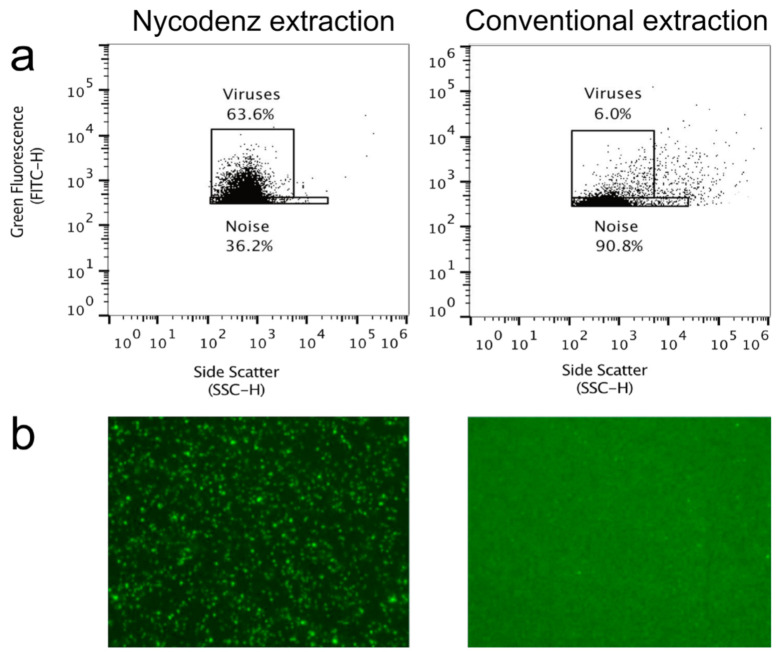
Example flow cytograms (**a**) and corresponding epifluorescence micrographs (**b**) of SYBR Green I stained virus particles extracted from sediment by the Nycodenz method (left) and conventional extraction method (right) [15]. Sediment was analyzed on the Attune NxT at JAMSTEC.

**Figure 3 viruses-13-00102-f003:**
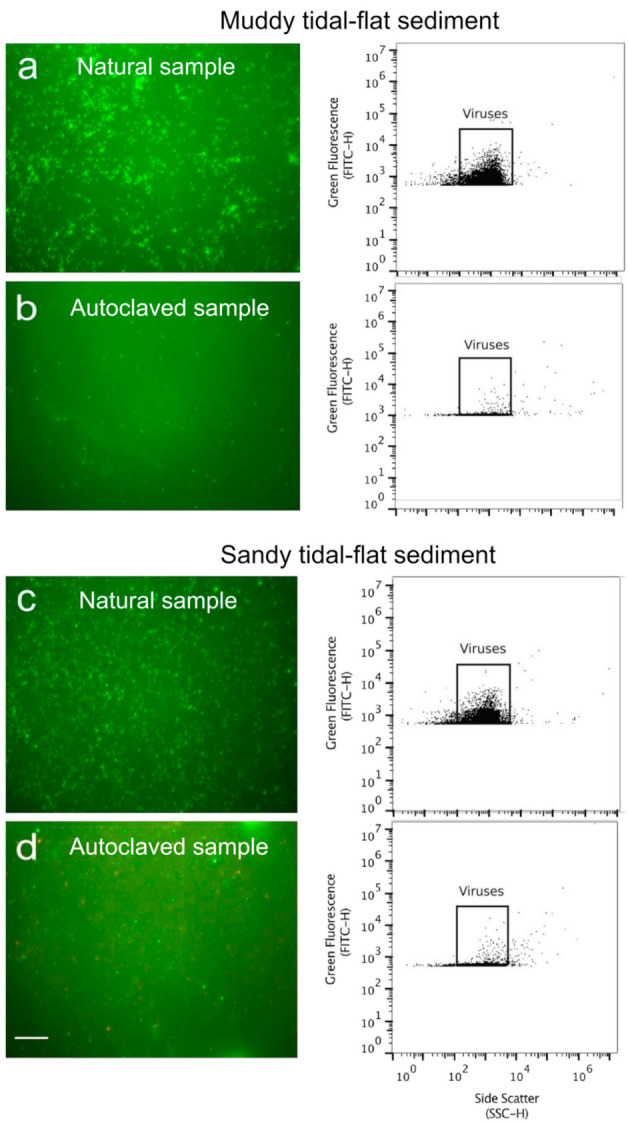
Epifluorescence micrographs and the corresponding cytograms of SYBR Green I stained virus particles from samples analyzed with epifluorescence microscopy (left column) and flow cytometry (right column). The samples were extracted from (**a**,**b**) muddy and (**c**,**d**) sandy tidal-flat sediment containing the natural virus population (**a**,**c**) and their autoclaved counterparts (**b**,**d**). All samples were prepared using the same dilution. Scale bar indicates 200 µm. Cytograms were recorded by the Accuri C6 flow cytometer. The gate used for quantifying the virus numbers is indicated by the square within the cytograms.

**Figure 4 viruses-13-00102-f004:**
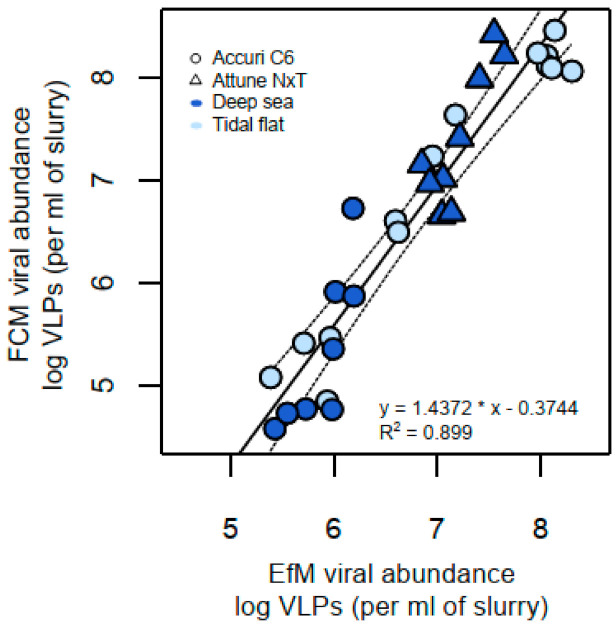
Regression of viral abundances enumerated from marine sediment samples (*n* = 31) by epifluorescence microscopy (EfM) and flow cytometry (FCM). Samples include deep-sea sediments (dark blue) and tidal-flat sediments (light blue). Results from the Accuri C6 flow cytometer are indicated by circles, and from the Attune NxT by triangles. The regression line (solid line) is shown along with 99% confidence intervals (dashed lines). Results are displayed in logarithmic scale.

**Figure 5 viruses-13-00102-f005:**
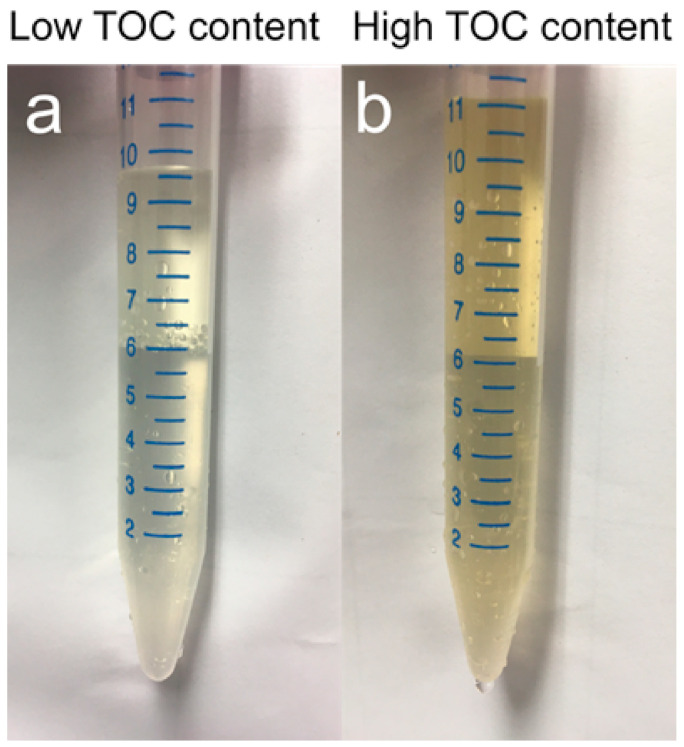
Comparison of two virus extracts derived from two different sediment samples with different total organic carbon (TOC) concentrations. TOC of 0.25% (**a**), estimated TOC 2–4% (**b**) [33].

**Table 1 viruses-13-00102-t001:** Characteristics of the marine sediment samples used to test the protocol for enumeration of sediment viruses via flow cytometry, for deep-sea sediments from 0 cm and 20 cm depth and tidal-flat sediments.

Sediment	Origin	Latitude	Longitude	Water Depth (m)	Chlorophyll A (mg/m^3^)	TOC (%) 0 cm 20 cm		Si/Al 0 cm 20 cm		Grain Size (%) >63 µm <63 µm	
Deep-sea	North Pacific Gyre	21°58′ N	178°19′ W	3250	49	0.65	0.52	3	3	-	-
Deep-sea	Equatorial upwelling	0°2′ S	179°59′ W	5285	102	0.63	0.25	3.7	3.4	-	-
Deep-sea	South Pacific Gyre	40°35′ S	179°15′ W	3089	56	0.64	0.64	3.7	3.6	-	-
Tidal-flat (Muddy)	German Wadden Sea	53°45′ N	8°13′ W	0	-	-	-	-	-	20.8	79.2
Tidal-flat (Sandy)	German Wadden Sea	53°45′ N	8°13′ W	0	-	-	-	-	-	60.1	39.8

**Table 2 viruses-13-00102-t002:** Linear regression analysis of viral abundances determined by flow cytometry and epifluorescence microscopy (from Figure 4). Regression analysis of individual results from both flow cytometers (Attune NxT and Accuri C6) and individual sediment types (deep sea, tidal flat, muddy, sandy) are included.

Sample Set	Slope	Intercept	R^2^	*n*	*p*-Value
All	1.4372	−0.3744	0.899	31	3.5 × 10^−16^
Attune NxT	1.9653	−0.8173	0.7092	9	2.7 × 10^−3^
Accuri C6	1.4377	−0.3813	0.9172	22	2.9 × 10^−12^
Deep sea	1.7017	−0.5881	0.8888	17	9.2 × 10^−9^
Tidal flat	1.2766	−0.2412	0.9355	14	1.0 × 10^−8^
Muddy tidal flat	1.1876	−0.1634	0.9338	7	2.5 × 10^−4^
Sandy tidal flat	1.3545	−0.3089	0.9323	7	2.6 × 10^−4^

## Data Availability

We do not report any data.

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
