# Peer review of "An Advanced Protocol for the Quantification of Marine Sediment Viruses via Flow Cytometry"

_viruses, 2021, doi:10.3390/v13010102_

Round 1

Reviewer 1 Report

The authors of the manuscript provide an advanced flow cytometry-based protocol for assessment of sediment viruses. Although the flow cytometry (FCM) technique has been used for estimation of aquatic viruses, the majority of the reports showed that the abundance estimated by FCM were higher than abundance counted by epifluorescence microscopy (EfM). In the present work, an advanced protocol which combines the use of a density gradient compound and flow cytometry is described in detail. A limitation of the study is the number of the sediment samples analysed. The results from both quantification methods (EfM and FCM) were highly correlated, but for some of the samples, viral abundances assessed by EfM exceeded FCM counts, especially in the lower abundance range. The developed protocol was tested on two different instruments in different research lab.

The development of analysis protocols based on flow cytometry could additionally contribute to the discrimination of viral population in aquatic ecosystems through the application of different of sensitive nucleic acid stains. Also, flow cytometry analysis in combination with viral sorting and single-virus sequencing will allow characterization of the viral diversity in aquatic ecosystems.

Author Response

The authors of the manuscript provide an advanced flow cytometry-based protocol for assessment of sediment viruses. Although the flow cytometry (FCM) technique has been used for estimation of aquatic viruses, the majority of the reports showed that the abundance estimated by FCM were higher than abundance counted by epifluorescence microscopy (EfM). In the present work, an advanced protocol which combines the use of a density gradient compound and flow cytometry is described in detail. A limitation of the study is the number of the sediment samples analysed. The results from both quantification methods (EfM and FCM) were highly correlated, but for some of the samples, viral abundances assessed by EfM exceeded FCM counts, especially in the lower abundance range. The developed protocol was tested on two different instruments in different research lab.

The development of analysis protocols based on flow cytometry could additionally contribute to the discrimination of viral population in aquatic ecosystems through the application of different of sensitive nucleic acid stains. Also, flow cytometry analysis in combination with viral sorting and single-virus sequencing will allow characterization of the viral diversity in aquatic ecosystems.

We thank Reviewer 1 for her/his positive feedback. We are aware that the number and diversity of samples was relatively low. However, as stated in the answers to reviewer two, we have slightly rephrased the manuscript. We now have clarified that we exclusively analyzed marine sediments from the deep sea and tidal flats. Even though, the sediments were spanning low and high organic carbon concentrations and virus abundances, we also removed the term “wide range of sediments”.

Reviewer 2 Report

Reading An advanced protocol for the quantification of sediment viruses via flow cytometry was a mix of excitement and a bit disheartening. There is definitely a spark of ingenuity here with some interesting ideas and concepts. However, there are a few things not addressed and missed within the study. One thing in particular, the authors need to do a complete check of their references because they are not quite correct in several places. For example, line 470 on page 13 has Mojica et al listed as [47] but in the ref list it is actually [46], and line 529 on page 14 lists refs for [52-54] but [54] is missing from the ref list. I'd highly recommend double checking all of them. In addition, this report studied marine sediments only, however the authors interchangeably use 'sediments' and 'all sediments' without actually have data to indicate that they examined all types of sediments, e.g. estuarine, freshwater or even sediments from more eutrophic environments. The following comments will be more specific:

Abstract, mentions marine sediments specifically then aquatic environments then just benthic viruses. The statement that enumeration of benthic viruses via FC using Nycodenz is limited since it could be interpreted as including every type of benthic virus. Please include the rest of your data from all locations if this is truly the case.

The first paragraph, introduction, lists reasons why it may be difficult to extract viruses from sediments. But the authors should include information regarding ionic bonding of viruses to those sediment particles. Please include it here and a ref.

Materials and methods, first and second paragraph discusses storage of samples prior to extraction of viruses. There is concern that the two types chosen actually may have contributed to loss. Did the authors flash freeze the samples to be stored at -80C? If not a slow freeze to thaw would cause issues. And were the samples thawed on ice? That would be critical. As for storage in 4C, it also isn't the optimal method. Please see Wen et al AEM 2004 70(7) p3862-3867 regarding accurate estimations of viral abundances via EfM. In addition, since the authors opted to sonicate the sediments, they need to cite Feliu et all Biotechnol Bioeng 58:536-540 and discuss the reason viral abundances may be higher due to lysing cells in the process.

Results, first paragraph has the unilateral use of 'sediment' for any benthic location, as this method was really more of an examination of two types of sediment not a wide range, please reword and restructure.

Table 1, probably a format error, but "Sediment" heading has the "t" below the rest of the word

Fig 2, please annotate on the figure above the columns of results which is Nycodenz and which is traditional (what do you mean by traditional as well). It is easier to grasp info if the figure has enough info to complete a picture for the reader.

Fig3, same comment as above, need annotations/labels indicating types or groups of samples.

Paragraph 3.2, page 10, lines 324-325, this is also misleading in the sense that this study only examined a few types of sediments, not a wide range. It just needs a little clarification perhaps indicating future research to discover if it will indeed cover a wide range.

Note: authors switch back and forth using "traditional" and "conventional" to refer to methods of extraction other than use of Nycodenz, please choose one.

Paragraph 4.2, page 12 line 436-437, would this be true for an environment with shallow sediments but highly eutrophic conditions as well?

Conclusions, page 13 first paragraph, since the author references aquatic studies of FCM counts, it would be interesting to include data from the overlying water of the sediment samples and compare this difference or similarity.

In Refs, the citations mix the use of abbreviating the author first initials with a space and without a space. See Ref 3 for an example where S. W. and C.A. are in the same line. A little thing to correct, nothing huge.

Author Response

Reading An advanced protocol for the quantification of sediment viruses via flow cytometry was a mix of excitement and a bit disheartening. There is definitely a spark of ingenuity here with some interesting ideas and concepts.

We thank Reviewer 2 for the overall positive remarks.

However, there are a few things not addressed and missed within the study. One thing in particular, the authors need to do a complete check of their references because they are not quite correct in several places. For example, line 470 on page 13 has Mojica et al listed as [47] but in the ref list it is actually [46], and line 529 on page 14 lists refs for [52-54] but [54] is missing from the ref list. I'd highly recommend double checking all of them.

We apologize for the inconsistent citations - never trust endnote. As recommended, we have carefully checked and corrected the citations and reference list.

In addition, this report studied marine sediments only, however the authors interchangeably use 'sediments' and 'all sediments' without actually have data to indicate that they examined all types of sediments, e.g. estuarine, freshwater or even sediments from more eutrophic environments.

Thank you for this remark. We agree and have now specified that we exclusively analyzed marine sediments from the deep sea and tidal flats and changed it throughout the manuscript. However, we think in some areas of the text, it may be safe to propose that since it works for marine sediments from different geographical areas and compositions, it might also work for other sediments.

The following comments will be more specific:

Abstract, mentions marine sediments specifically then aquatic environments then just benthic viruses. The statement that enumeration of benthic viruses via FC using Nycodenz is limited since it could be interpreted as including every type of benthic virus. Please include the rest of your data from all locations if this is truly the case.

We mentioned “aquatic environments” to set up our investigation in the context of past viral flow cytometry work (which has been nearly entirely in aquatic environments). We had previously referred to “sediment viruses” synonymously with “benthic viruses”. However, to avoid confusion, we now have unified the terms and use “sediment viruses” exclusively and specified that they are marine sediments (as mentioned in the previous response). We also rephrased parts of the abstract to make clear that we only used marine sediment types and changed the title accordingly.

The first paragraph, introduction, lists reasons why it may be difficult to extract viruses from sediments. But the authors should include information regarding ionic bonding of viruses to those sediment particles. Please include it here and a ref.

Thank you for this remark. We have added a paragraph and references accordingly (lines 78ff.).

Materials and methods, first and second paragraph discusses storage of samples prior to extraction of viruses. There is concern that the two types chosen actually may have contributed to loss. Did the authors flash freeze the samples to be stored at -80C? If not a slow freeze to thaw would cause issues. And were the samples thawed on ice? That would be critical.

All deep-sea samples were indeed flash frozen in liquid nitrogen, immediately stored at -80°C and thawed on ice. This is now indicated in the methods section (lines 127ff.).

As for storage in 4C, it also isn't the optimal method. Please see Wen et al AEM 2004 70(7) p3862-3867 regarding accurate estimations of viral abundances via EfM.

The tidal-flat sediments were stored at 4°C before extraction. While not optimal, as this was roughly the average in-situ temperature at the sampling time point, we intended not to change the conditions until processing. Furthermore, the focus of this manuscript was not to present natural viral abundances, but rather, to compare the performance of the new proposed flow cytometry method with standard microscopy counts. For both methods, samples were subjected to the same conditions.

In addition, since the authors opted to sonicate the sediments, they need to cite Feliu et all Biotechnol Bioeng 58:536-540 and discuss the reason viral abundances may be higher due to lysing cells in the process.

This is an important remark. However, since sonication has already been validated in previous sediment virus methods (e.g. Danovaro et al., 2001; Danovaro and Middleboe, 2010), we did not change anything here. We found that our results are in accordance with previous investigations on tidal-flat sediments and South Pacific Gyre sediments (lines 394-402), but again, the focus of this manuscript is on evaluating the performance of the proposed flow cytometry method rather than presenting natural viral abundances.

Results, first paragraph has the unilateral use of 'sediment' for any benthic location, as this method was really more of an examination of two types of sediment not a wide range, please reword and restructure.

Done as suggested throughout the manuscript.

Table 1, probably a format error, but "Sediment" heading has the "t" below the rest of the word

This was indeed a formatting error. Fixed.

Fig 2, please annotate on the figure above the columns of results which is Nycodenz and which is traditional (what do you mean by traditional as well). It is easier to grasp info if the figure has enough info to complete a picture for the reader.

Fig3, same comment as above, need annotations/labels indicating types or groups of samples.

Thank you for this suggestion. We changed the figures accordingly.

Paragraph 3.2, page 10, lines 324-325, this is also misleading in the sense that this study only examined a few types of sediments, not a wide range. It just needs a little clarification perhaps indicating future research to discover if it will indeed cover a wide range.

Done as suggested. In the conclusion section (lines 557ff.), a statement regarding future research was also added.

Note: authors switch back and forth using "traditional" and "conventional" to refer to methods of extraction other than use of Nycodenz, please choose one.

We have chosen “conventional” and use it throughout the manuscript.

Paragraph 4.2, page 12 line 436-437, would this be true for an environment with shallow sediments but highly eutrophic conditions as well?

You are right. We now included “highly eutrophic sediments” in the sentence below (lines 459ff.): “Samples with high clay, humic content, or high organic matter, such as highly eutrophic sediments, may benefit from extractions using smaller sediment volumes, e.g. by higher dilutions of slurries [22, 42].”

Conclusions, page 13 first paragraph, since the author references aquatic studies of FCM counts, it would be interesting to include data from the overlying water of the sediment samples and compare this difference or similarity.

Good suggestion. We would also be interested in knowing, but unfortunately, we do not have these samples.

In Refs, the citations mix the use of abbreviating the author first initials with a space and without a space. See Ref 3 for an example where S. W. and C.A. are in the same line. A little thing to correct, nothing huge.

Thank you again for carefully reading our manuscript. We hope we could eliminate all of these little inaccuracies.